# Synthesis of Novel 1,3,4-Oxadiazole-Derived *α*-Aminophosphonates/*α*-Aminophosphonic Acids and Evaluation of Their In Vitro Antiviral Activity against the Avian Coronavirus Infectious Bronchitis Virus

**DOI:** 10.3390/pharmaceutics15010114

**Published:** 2022-12-29

**Authors:** Shaima Hkiri, Marwa Mekni-Toujani, Elvan Üstün, Karim Hosni, Abdeljelil Ghram, Soufiane Touil, Ali Samarat, David Sémeril

**Affiliations:** 1Synthèse Organométallique et Catalyse, UMR-CNRS 7177, University of Strasbourg, 4 rue Blaise Pascal, 67008 Strasbourg, France; 2Laboratory of Hetero-Organic Compounds and Nanostructured Materials, LR18ES11, Faculty of Sciences of Bizerte, University of Carthage, Bizerte 7021, Tunisia; 3Laboratory of Epidemiology and Veterinary Microbiology, LR19IP03, Institute Pasteur of Tunis, University of Tunis El Manar, BP 74, Tunis-Belvedere 1002, Tunisia; 4Department of Chemistry, Faculty of Science and Arts, University of Ordu, 52200 Ordu, Turkey; 5Laboratoire des Substances Naturelles, Institut National de Recherche et d’Analyse Physico-Chimique (INRAP), Biotechpôle de Sidi Thabet, Ariana 2020, Tunisia

**Keywords:** *α*-aminophosphonate, *α*-aminophosphonic acid, 1,3,4-oxadiazole, infectious bronchitis virus, virucidal activity, coronavirus

## Abstract

An efficient and simple approach has been developed for the synthesis of eight dialkyl/aryl[(5-phenyl-1,3,4-oxadiazol-2-ylamino)(aryl)methyl]phosphonates through the Pudovik-type reaction of dialkyl/arylphosphite with imines, obtained from 5-phenyl-1,3,4-oxadiazol-2-amine and aromatic aldehydes, under microwave irradiation. Five of them were hydrolyzed to lead to the corresponding phosphonic acids. Selected synthesized compounds were screened for their in vitro antiviral activity against the avian bronchitis virus (IBV). In the MTT cytotoxicity assay, the dose-response curve showed that all test compounds were safe in the range concentration of 540–1599 µM. The direct contact of novel synthesized compounds with IBV showed that the diethyl[(5-phenyl-1,3,4-oxadiazol-2-ylamino)(4-trifluoromethoxyphenyl)methyl]phosphonate (**5f**) (at 33 µM) and the [(5-phenyl-1,3,4-oxadiazol-2-ylamino)(4-trifluoromethylphenyl)methyl] phosphonic acid (**6a**) (at 1.23 µM) strongly inhibited the IBV infectivity, indicating their high virucidal activity. However, virus titers from IBV-infected Vero cells remained unchanged in response to treatment with the lowest non-cytotoxic concentrations of synthesized compounds suggesting their incapacity to inhibit the virus replication inside the host cell. Lack of antiviral activity might presumably be ascribed to their polarity that hampers their diffusion across the lipophilic cytoplasmic membrane. Therefore, the interactions of **5f** and **6a** were analyzed against the main coronavirus protease, papain-like protease, and nucleocapsid protein by molecular docking methods. Nevertheless, the novel 1,3,4-oxadiazole-based *α*-aminophosphonic acids and *α*-amino-phosphonates hold potential for developing new hygienic virucidal products for domestic, chemical, and medical uses.

## 1. Introduction

Coronaviruses (Coronaviridae family) are highly infectious RNA viruses that affect the respiratory, gastrointestinal, urinary, and neurological systems of mammals and birds [1,2,3,4]. Depending on their genome, coronaviruses (CoVs) are divided into four groups: alpha-, beta-, gamma-, and delta-coronaviruses [5]. The alpha and beta-coronaviruses are particularly worrisome because they can cross the species barriers and infect humans, thus causing diseases ranging from upper respiratory tract infections to severe acute respiratory syndrome (SARS) [6].

Avian infectious bronchitis virus (IBV), a widespread gamma-coronavirus that infected both wild birds, such as in Mont-Saint-Michel bay last summer, and breeding birds, such as chickens. IBV is one of the foremost causes of economic losses to the poultry industry as a result of a disturbance in feed conversion and weight gain with a concomitant reduction in egg production and quality and, in some cases, increased chicken mortality [7]. As for SARS-CoV-2, IBV exhibited high genomic diversity and elevated mutation frequency, leading to different virus serotypes that complicate the control of IBV and widen its host range [6]. The latter biological property is determined by the spike glycoprotein (a key determinant of virus pathogenicity), which allows a virus to bind to different target cells, including respiratory (nose, trachea, lung, and pulmonary alveoli), kidneys, oviduct, testes, and alimentary (esophagus, proventriculus, duodenum, jejunum, bursa of Fabricius, rectum, and cloaca) [6,8,9].

Vaccination using a live vaccine, inactivated virus, subunit vaccine, and vector vaccine represents the main way of controlling IBV, despite its inefficiency in managing some variant serotypes [6,7]. Therefore, searching for alternative and efficient treatment for IBV control is a challenging task. Recent efforts in this direction have successfully identified some putative anti-IBV, including natural-like plant-derived oleoresins, essential oils, and extracts (i.e., extracts from Mentha piperita, Melissa officinalis, Thymus vulgaris, Hyssopus officinalis, Salvia officinalis) [7,10,11]. Alternatively, synthetic chemicals (i.e., lithium chloride) have also been applied as potent anti-IBV [12].

As bioisosteres [13] of amino acids, α-aminophosphonates and their *α*-aminophosphonic acid derivatives have attracted wide attention in medicinal chemistry and agrochemical sciences. Indeed, the tetrahedral conformation of the phosphorus atom allows it to be used in place of a carbon atom, which makes it possible to obtain analogous transition states [14] with the receptors to which amino acids normally bind. Taking into account their broad spectrum of biological activities, *α*-aminophosphonates and *α*-aminophosphonic acids have been the subjects of intense research in recent years [15,16]. Indeed, these compounds are known as enzyme inhibitors [17,18,19,20], anti-human immunodeficiency virus (anti-HIV) [21], anticancer [22,23,24,25], antibacterial [26,27], and herbicides [28,29,30], of which glyphosate (Round up^®^) and Dufulin (Figure 1) are known the most.

Recently, drugs containing a 1,3,4-oxadiazole ring have emerged as promising antiviral agents [31,32,33] and some of them are commercialized, such as anti-HIV (Raltegravir) or antibiotic (Furamizole) (Figure 1).

In this context and in connection with our research projects on the synthesis of novel phosphorinated compounds with possible biological properties [34,35], we now report an efficient, fast and simple way to obtain 1,3,4-oxadiazole-derived *α*-aminophosphonates/*α*-aminophosphonic acids. A selection of the synthesized compounds was tested for their possible antiviral activity against the IBV virus. The activity details of the two most active compounds were analyzed by molecular docking methods against coronavirus main proteinase, papain-like protease, and N/C-terminal domain of coronavirus nucleocapsid protein.

## 2. Materials and Methods

### 2.1. Apparatus, Materials, and Analysis

All manipulations involving phosphorus derivatives were carried out under dry argon. Solvents were dried by conventional methods and were distilled immediately before use. Routine ^1^H, ^13^C{^1^H}, ^31^P{^1^H}, and ^19^F{^1^H} spectra were recorded with Bruker FT instruments (AC 300 and 500). ^1^H NMR spectra were referenced to residual protonated solvents (*δ* = 2.50 ppm for DMSO-d_6_). ^13^C NMR chemical shifts are reported relative to deuterated solvents (*δ* = 39.5 ppm for DMSO-d_6_). ^31^P and ^19^F NMR spectroscopic data are given relative to external H_3_PO_4_ and CCl_3_F, respectively. Chemical shifts and coupling constants are reported in ppm and Hz, respectively. Microwave irradiation was carried out using the CEM Discover microwave synthesis system equipped with a pressure controller (17 bar). Melting points were determined with a Büchi 535 capillary melting point apparatus. Infrared spectra were recorded on a Bruker FT-IR Alpha-P spectrometer. Elemental analyses were carried out by the Service de Microanalyse, Institut de Chimie, Université de Strasbourg. 5-Phenyl-1,3,4-oxadiazol-2-amine (**1**) [36], diethyl[(5-phenyl-1,3,4-oxadiazol-2-ylamino)(4-trifluoromethylphenyl)methyl]phosphona-te (**5a**) [37], diethyl[(5-phenyl-1,3,4-oxadiazol-2-ylamino)(4-nitrophenyl)methyl]phos- phonate (**5b**) [37], and diethyl[(5-phenyl-1,3,4-oxadiazol-2-ylamino)(4-fluorophenyl) methyl]phosphonate (**5c**) [38] were prepared by literature procedures.

### 2.2. General Procedure for the Synthesis of (E)-1-(Aryl)-N-(5-Phenyl-1,3,4-Oxadiazol-2-yl) Methanimine

In a round-bottomed flask equipped with a Dean-Stark apparatus, a mixture of phenyl-1,3,4-oxadiazol-2-amine (**1**) (2.0 mmol) and aryl aldehyde (3.0 mmol) in toluene (20 mL) was refluxed for 48 h. After cooling to room temperature, the solvent was evaporated, and the crude product was washed with cool ethanol (15 mL), filtered, and dried under a vacuum to afford the desired imine.

(*E*)-1-(3,4,5-Trifluophenyl)-*N*-(5-phenyl-1,3,4-oxadiazol-2-yl)methanimine (**2d**). White solid, yield 89%. ^1^H NMR (300 MHz, DMSO-d_6_): *δ* = 9.34 (s, 1H, N=CH), 8.11–8.04 (m, 4H, arom. CH), 7.69–7.60 (m, 3H, arom. CH) ppm; ^13^C{^1^H} NMR (126 MHz, DMSO-d_6_): *δ* = 167.2 (s, N=CH), 165.5 (s, arom. Cquat C(N)=N), 163.1 (s, arom. Cquat C(Ph)=N), 150.7 (dd, arom. Cquat meta CF, ^1^*J*_CF_ = 250.6 Hz, ^2^*J*_CF_ = 10.1 Hz), 142.4 (dt, arom. Cquat para CF, ^1^*J*_CF_ = 258.8 Hz, ^2^*J*_CF_ = 14.1 Hz), 132.2 (s, arom. CH), 130.9 (s, arom. Cquat of C_6_H_2_F_3_), 129.5 (s, arom. CH), 126.4 (s, arom. CH), 123.4 (s, arom. Cquat of C_6_H_5_), 114.7 (d, arom. CH, ^2^*J*_CF_ = 21.7 Hz) ppm; ^19^F{^1^H} NMR (282 MHz, DMSO-d_6_): *δ* = -133.1 (d, meta CF of C_6_H_2_F_3_, ^3^*J*_FF_ = 20.6 Hz), −152.2 (t, para CF of C_6_H_2_F_3_, ^3^*J*_FF_ = 20.6 Hz) ppm. IR: *ν* = 1650 (C=NH), 1614 (C=N) cm^−1^. Elemental analysis calcd. (%) for C_15_H_8_ON_3_F_3_ (303.24): C 59.41, H 2.66, N 13.86; found C 59.45, H 2.69 N, 13.81.

(*E*)-1-(3,5-Bis-trifuoromethyl-phenyl)-*N*-(5-phenyl-1,3,4-oxadiazol-2-yl)methan- imine (**2e**). White solid, yield 73%. ^1^H NMR (300 MHz, DMSO-d_6_): *δ* = 9.58 (s, 1H, N=CH), 8.75 (s, 2H, arom. CH of C_6_H_3_(CF_3_)_2_), 8.47 (s, 1H, arom. CH of C_6_H_3_(CF_3_)_2_), 8.08 (d, 2H, arom. CH of C_6_H_5_, ^3^*J*_HH_ = 6.0 Hz), 7.69–7.63 (m, 3H, arom. CH of C_6_H_5_) ppm; ^13^C{^1^H} NMR (126 MHz, DMSO-d_6_): *δ* = 167.6 (s, N=CH), 165.5 (s, arom. Cquat C(N)=N), 163.2 (s, arom. Cquat C(Ph)=N), 136.6 (s, arom. Cquat of C_6_H_3_(CF_3_)_2_), 132.3 (s, arom. CH), 131.2 (q, arom. Cquat CCF_3_, ^2^*J*_CF_ = 33.5 Hz), 130.2 (s, arom. CH), 129.5 (s, arom. CH), 126.8 (s, arom. CH), 126.5 (s, arom. CH), 123.3 (s, arom. Cquat of C_6_H_5_), 123.0 (q, CF_3_, ^1^*J*_CF_ = 273.4 Hz) ppm; ^19^F{^1^H} NMR (282 MHz, DMSO-d_6_): *δ* = −61.5 (s, CF_3_) ppm. IR: *ν* = 1645 (C=NH), 1615 (C=N) cm^−1^. Elemental analysis calcd. (%) for C_17_H_9_ON_3_F_6_ (385.26): C 53.00, H 2.35, N 10.91; found C 52.86, H 2.41, N 10.88.

(*E*)-1-(4-Trifluoromethoxyphenyl)-*N*-(5-phenyl-1,3,4-oxadiazol-2-yl)methanimine (**2f**). White solid, yield 83%. ^1^H NMR (300 MHz, DMSO-d_6_): *δ* = 9.40 (s, 1H, N=CH), 8.25 (d, 2H, arom. CH of C_6_H_4_OCF_3_, ^3^*J*_HH_ = 9.0 Hz), 8.09–8.06 (m, 2H, arom. CH), 7.66–7.59 (m, 5H, arom. CH) ppm; ^13^C{^1^H} NMR (126 MHz, DMSO-d_6_): *δ* = 168.5 (s, N=CH), 166.0 (s, arom. Cquat C(N)=N), 162.9 (s, arom. Cquat C(Ph)=N), 152.1 (s, arom. Cquat COCF_3_), 133.28 (s, arom. Cquat of C_6_H_4_OCF_3_), 132.5 (s, arom. CH), 131.1 (s, arom. CH), 129.5 (s, arom. CH), 126.4 (s, arom. CH), 123.5 (s, arom. Cquat of C_6_H_5_), 121.4 (s, arom. CH), 119.9 (q, OCF_3_, ^1^*J*_CF_ = 258.4 Hz) ppm; ^19^F{^1^H} NMR (282 MHz, DMSO-d_6_): *δ* = −56.6 (s, OCF_3_) ppm. IR: *ν* = 1650 (C=NH), 1612 (C=N) cm^−1^. Elemental analysis calcd. (%) for C_16_H_10_O_2_N_3_F_3_ (333.27): C 57.66, H 3.02, N 12.61; found C 57.48, H 2.99, N 12.46.

### 2.3. General Procedure for the Synthesis of Dialkyl/aryl[(5-Phenyl-1,3,4-Oxadiazol-2-Ylamino) (Aryl)Methyl]phosphonate

In a round-bottomed flask, a mixture of (*E*)-1-(aryl)-*N*-(5-phenyl-1,3,4-oxadiazol- 2-yl)methanimine (1.0 mmol) and dialkyl/arylphosphite (2.0 mmol) was added and irradiated under microwave in neat condition at 115 °C for 10 min at 300 W. After cooling to room temperature, the crude product was washed with Et_2_O (3 × 10 mL), filtered, and dried under vacuum.

Dimethyl[(5-phenyl-1,3,4-oxadiazol-2-ylamino)(4-trifluoromethylphenyl)methyl] phosphonate (**3**). Yield 54%. ^1^H NMR (300 MHz, DMSO-d_6_): *δ* = 9.17 (dd, ^1^H, NH, ^3^*J*_HH_ = 9.6 Hz, ^3^*J*_PH_ = 3.6 Hz), 7.84–7.78 (m, 6H, arom. CH), 7.56–7.53 (m, 3H, arom. CH), 5.46 (dd, 1H, CHP, ^2^*J*_PH_ = 22.5 Hz, ^3^*J*_HH_ = 9.6 Hz), 3.70 (d, 3H, OCH_3_, ^3^*J*_PH_ = 10.5 Hz), 3.61 (d, 3H, OCH_3_, ^3^*J*_PH_ = 10.8 Hz) ppm; ^13^C{^1^H} NMR (126 MHz, DMSO-d_6_): *δ* = 162.9 (d, arom. Cquat para of CF_3_, ^2^*J*_CP_ = 11.5 Hz), 158.5 (s, arom. Cquat C(Ph)=N), 140.3 (s, arom. Cquat C(NH)=N), 130.8 (s, arom. CH), 129.3 (s, arom. CH), 128.9 (d, arom. CH, ^3^*J*_CP_ = 4.8 Hz), 128.5 (q, arom. Cquat CCF_3_, ^2^*J*_CF_ = 32.4 Hz), 125.3 (s, arom. CH), 125.3 (s, arom. CH), 124.2 (q, CF_3_, ^1^*J*_CF_ = 272.8 Hz), 123.9 (s, arom. Cquat of C_6_H_5_), 53.7 (d, OCH_3_, ^2^*J*_CP_ = 25.7 Hz), 53.6 (d, OCH_3_, ^2^*J*_CP_ = 25.7 Hz), 53.5 (d, CHP, ^1^*J*_CP_ = 153.5 Hz) ppm; ^31^P{^1^H} NMR (121 MHz, DMSO-d_6_): *δ* = 21.7 (q, P(O), ^7^*J*_PF_ = 2.3 Hz) ppm; ^19^F{^1^H} NMR (282 MHz, DMSO-d_6_): *δ* = −61.0 (d, CF_3_, ^7^*J*_PF_ = 2.3 Hz) ppm. IR: *ν* = 1618 (C=N) cm^−1^. Elemental analysis calcd. (%) for C_18_H_17_O_4_N_3_F_3_P (427.31): C 50.59, H 4.01, N 9.83; found C 50.48, H 3.94, N 9.75.

Diphenyl[(5-phenyl-1,3,4-oxadiazol-2-ylamino)(4-trifluoromethylphenyl)methyl] phosphonate (**4**). Yield 90%. ^1^H NMR (300 MHz, DMSO-d_6_): *δ* = 9.55 (dd, 1H, NH, ^3^*J*_HH_ = 10.2 Hz, ^3^*J*_PH_ = 3.0 Hz), 7.97 (d, 2H, arom. CH, ^3^*J*_HH_ = 7.5 Hz), 7.86–7.81 (m, 4H, arom. CH), 7.56–7.54 (m, 3H, arom. CH), 7.38–7.31 (m, 4H, arom. CH), 7.19 (t, 1H, arom. CH, ^3^*J*_HH_ = 7.4 Hz), 7.19 (t, 1H, arom. CH, ^3^*J*_HH_ = 7.4 Hz), 7.10 (d, 2H, arom. CH, ^3^*J*_HH_ = 8.4 Hz), 7.02 (d, 2H, arom. CH, ^3^*J*_HH_ = 8.4 Hz), 5.95 (dd, 1H, CHP, ^2^*J*_PH_ = 23.1 Hz, ^3^*J*_HH_ = 10.0 Hz) ppm; ^13^C{^1^H} NMR (126 MHz, DMSO-d_6_): *δ* = 162.8 (d, arom. Cquat para of CF_3_, ^2^*J*_CP_ = 11.3 Hz), 158.7 (s, arom. Cquat C(Ph)=N), 149.9 (d, arom. Cquat of OC_6_H_5_, ^2^*J*_CP_ = 9.7 Hz), 149.7 (d, arom. Cquat of OC_6_H_5_, ^2^*J*_CP_ = 9.2 Hz), 139.0 (s, arom. Cquat C(NH)=N), 130.9 (s, arom. CH), 129.9 (s, arom. CH), 129.3 (s, arom. CH), 129.0 (q, arom. Cquat CCF_3_, ^2^*J*_CF_ = 33.4 Hz), 125.5 (s, arom. CH), 125.5 (s, arom. CH), 125.5 (s, arom. CH), 125.4 (s, arom. CH), 124.1 (q, CF_3_, ^1^*J*_CF_ = 272.5 Hz), 123.0 (s, arom. Cquat of C_6_H_5_), 120.3 (s, arom. CH), 120.3 (s, arom. CH), 54.6 (d, CHP, ^1^*J*_CP_ = 157.1 Hz) ppm; ^31^P{^1^H} NMR (121 MHz, DMSO-d_6_): *δ* = 12.6 (q, P(O), ^7^*J*_PF_ = 2.4 Hz) ppm; ^19^F{^1^H} NMR (282 MHz, DMSO-d_6_): *δ* = −61.1 (d, CF_3_, ^7^*J*_PF_ = 2.4 Hz) ppm. IR: *ν* = 1615 (C=N) cm^−1^. Elemental analysis calcd. (%) for C_28_H_21_O_4_N_3_F_3_P (551.45): C 60.98, H 3.84, N 7.62; found C 61.03, H 3.88, N 7.55.

Diethyl[(5-phenyl-1,3,4-oxadiazol-2-ylamino)(3,4,5-trifluophenyl)methyl]phospho- nate (**5d**). White solid, yield 90%. ^1^H NMR (300 MHz, DMSO-d_6_): *δ* = 9.00 (dd, 1H, NH, ^3^*J*_HH_ = 6.0 Hz, ^3^*J*_PH_ = 1.8 Hz), 7.85–7.83 (m, 2H, arom. CH), 7.59–7.53 (m, 5H, arom. CH), 5.38 (dd, 1H, CHP, ^2^*J*_PH_ = 13.2 Hz, ^3^*J*_HH_ = 6.0 Hz), 4.10–3.89 (m, 4H, OC*H*_2_CH_3_), 1.18 (t, 3H, OCH_2_C*H*_3_, ^3^*J*_HH_ = 4.2 Hz), 1.12 (t, 3H, OCH_2_C*H*_3_, ^3^*J*_HH_ = 4.2 Hz) ppm; ^13^C{^1^H} NMR (126 MHz, DMSO-d_6_): *δ* = 162.8 (d, arom. Cquat of C_6_H_2_F_3_, ^2^*J*_CP_ = 10.7 Hz), 158.5 (s, arom. Cquat C(Ph)=N), 149.9 (dd, arom. Cquat CF, ^1^*J*_CF_ = 247.5 Hz, ^2^*J*_CF_ = 9.2 Hz), 138.3 (dt, arom. Cquat CF, ^1^*J*_CF_ = 250.0 Hz, ^2^*J*_CF_ = 14.4 Hz), 133.0 (d, arom. Cquat C(NH)=N, ^3^*J*_CP_ = 4.0 Hz), 130.8 (s, arom. CH), 129.3 (s, arom. CH), 125.3 (s, arom. CH), 123.9 (s, arom. Cquat of C_6_H_5_), 112.9 (dd, arom. CH of C_6_H_2_F_3_, ^2^*J*_CF_ = 17.5 Hz, ^3^*J*_CF_ = 4.5 Hz), 63.0 (d, *C*H_2_CH_3_, ^2^*J*_CP_ = 19.1 Hz), 63.0 (d, *C*H_2_CH_3_, ^2^*J*_CP_ = 18.9 Hz), 53.2 (d, CHP, ^1^*J*_CP_ = 154.7 Hz), 16.2 (d, CH_2_*C*H_3_, ^3^*J*_CP_ = 14.1 Hz), 16.1 (d, CH_2_*C*H_3_, ^3^*J*_CP_ = 14.4 Hz) ppm; ^31^P{^1^H} NMR (121MHz, DMSO-d_6_): *δ* = 18.9 (dt, P(O), ^6^*J*_PF_ = 4.8 Hz, ^5^*J*_PF_ = 1.4 Hz) ppm; ^19^F{^1^H} NMR (282 MHz, DMSO-d_6_): *δ* = −135.1 (dd, meta CF of C_6_H_2_F_3_, ^3^*J*_FF_ = 21.7 Hz, ^5^*J*_PF_ = 1.2 Hz), −162.0 (td, para CF of C_6_H_2_F_3_, ^3^*J*_FF_ = 21.7 Hz, ^6^*J*_PF_ = 5.0 Hz) ppm. IR: *ν* = 1598 (C=N) cm^−1^. Elemental analysis calcd. (%) for C_19_H_18_O_4_N_3_F_3_P (440.33): C 51.82, H 4.12, N 9.54; found C 51.78, H 4.07, N 9.41.

Diethyl[(5-phenyl-1,3,4-oxadiazol-2-ylamino)(3,5-bis-trifuoromethylphenyl)methyl] phosphonate (**5e**). White solid, yield 97%. ^1^H NMR (300 MHz, DMSO-d_6_): *δ* = 9.18 (dd, 1H, NH, ^3^*J*_HH_ = 10.2 Hz, ^3^*J*_PH_ = 3.3 Hz), 8.36 (s, 2H, arom. CH of C_6_H_3_(CF_3_)_2_), 8.10 (s, 1H, arom. CH of C_6_H_3_(CF_3_)_2_), 7.87–7.84 (m, 2H, arom. CH), 7.55–7.53 (m, 3H, arom. CH), 5.69 (dd, 1H, CHP, ^2^*J*_PH_ = 22.2 Hz, ^3^*J*_HH_ = 10.2 Hz), 4.12–3.90 (m, 4H, OC*H*_2_CH_3_), 1.16 (t, 3H, OCH_2_C*H*_3_, ^3^*J*_HH_ = 7.0 Hz), 1.08 (t, 3H, OCH_2_C*H*_3_, ^3^*J*_HH_ = 7.0 Hz) ppm; ^13^C{^1^H} NMR (126 MHz, DMSO-d_6_): *δ* = 162.7 (d, arom. Cquat of C_6_H_3_(CF_3_)_2_, ^2^*J*_CP_ = 9.7 Hz), 158.5 (s, arom. Cquat C(Ph)=N), 139.3 (s, arom. Cquat C(NH)=N), 130.9 (s, arom. CH), 130.1 (q, arom. Cquat CCF_3_, ^2^*J*_CF_ = 32.8 Hz), 129.3 (s, arom. CH), 129.1 (s, arom. CH), 125.3 (s, arom. CH), 123.3 (q, CF_3_, ^1^*J*_CF_ = 273.4 Hz), 123.9 (s, arom. Cquat of C_6_H_5_), 121.8 (s, arom. CH), 63.1 (d, O*C*H_2_CH_3_, ^2^*J*_CP_ = 30.4 Hz), 63.0 (d, O*C*H_2_CH_3_, ^2^*J*_CP_ = 30.4 Hz), 53.4 (d, CHP, ^1^*J*_CP_ = 152.8 Hz), 16.0 (d, OCH_2_*C*H_3_, ^3^*J*_CP_ = 22.5 Hz), 16.0 (d, OCH_2_*C*H_3_, ^3^*J*_CP_ = 22.7 Hz) ppm; ^31^P{^1^H} NMR (121 MHz, DMSO-d_6_): *δ* = 18.6 (s, P(O)) ppm; ^19^F{^1^H} NMR (282 MHz, DMSO-d_6_): *δ* = −61.3 (s, CF_3_) ppm. IR: *ν* = 1608 (C=N) cm^−1^. Elemental analysis calcd. (%) for C_21_H_20_O_4_N_3_F_6_P (523.37): C 48.19, H 3.85, N 8.03; found C 48.35, H 4.01, N 7.92.

Diethyl[(5-phenyl-1,3,4-oxadiazol-2-ylamino)(4-trifluoromethoxyphenyl)methyl] phosphonate (**5f**). White solid, yield 92%. M.p. 106.6–107.5 °C; ^1^H NMR (300 MHz, DMSO-d_6_): *δ* = 9.06 (dd, ^1^H, NH, ^3^*J*_HH_ = 9.9 Hz, ^3^*J*_PH_ = 3.6 Hz), 7.85–7.81 (m, 2H, arom. CH), 7.69 (dd, 2H, arom. CH, ^3^*J*_HH_ = 8.4 Hz, ^4^*J*_HH_ = 1.8 Hz), 7.56–7.51 (m, 3H, arom. CH), 7.40 (d, 2H, arom. CH, ^3^*J*_HH_ = 8.4 Hz), 5.30 (dd, 1H, CHP, ^2^*J*_PH_ = 21.9 Hz, ^3^*J*_HH_ = 9.9 Hz), 4.11–3.82 (m, 4H, OC*H*_2_CH_3_), 1.17 (t, 3H, OCH_2_C*H*_3_, ^3^*J*_HH_ = 7.5 Hz), 1.08 (t, 3H, OCH_2_C*H*_3_, ^3^*J*_HH_ = 7.1 Hz) ppm; ^13^C{^1^H} NMR (126 MHz, DMSO-d_6_): *δ* = 163.0 (d, arom. Cquat para of OCF_3_, ^2^*J*_CP_ = 11.0 Hz), 158.4 (s, arom. Cquat C(Ph)=N), 148.0 (s, arom. Cquat C(NH)=N), 135.0 (s, arom. Cquat COCF_3_), 130.8 (s, arom. CH), 130.1 (d, arom. CH, ^3^*J*_CP_ = 4.8 Hz), 129.3 (s, arom. CH), 125.3 (s, arom. CH), 123.1 (s, arom. Cquat of C_6_H_5_), 120.9 (s, arom. CH), 120.1 (q, OCF_3_, ^1^*J*_CF_ = 256.8 Hz), 62.8 (d, O*C*H_2_CH_3_, ^2^*J*_CP_ = 18.8 Hz), 62.8 (d, O*C*H_2_CH_3_, ^2^*J*_CP_ = 18.5 Hz), 53.7 (d, CHP, ^1^*J*_CP_ = 154.7 Hz), 16.2 (d, OCH_2_*C*H_3_, ^3^*J*_CP_ = 5.0 Hz), 16.1 (d, OCH_2_*C*H_3_, ^3^*J*_CP_ = 5.1 Hz) ppm; ^31^P{^1^H} NMR (121 MHz, DMSO-d_6_): *δ* = 19.7 (s, P(O)) ppm; ^19^F{^1^H} NMR (282 MHz, DMSO-d_6_): *δ* = −56.8 (s, OCF_3_) ppm. IR: *ν* = 1607 (C=N) cm^−1^. Elemental analysis calcd. (%) for C_20_H_21_O_5_N_3_F_3_P (471.37): C 50.96, H 4.49, N 8.91; found C 50.83, H 4.37, N 8.86.

### 2.4. General Procedure for the Synthesis of [(5-Phenyl-1,3,4-Oxadiazol-2-Ylamino)(Aryl) Methyl]Phosphonic acids

In a Schlenk tube, a mixture of diethyl[(5-phenyl-1,3,4-oxadiazol-2-ylamino)(aryl) methyl]phosphonate (0.5 mmol) and trimethylsilyl bromide (0.53 mL, 4.0 mmol) were dissolved in CH_2_Cl_2_ (5 mL). The reaction mixture was stirred at room temperature. After 72 h, the reaction mixture was evaporated, and the crude product was dissolved into a solution of NaOH 2 M (10 mL) and filtrated. The white product was precipitated from the solution by the addition of HCl 1 M (25 mL). The white solid was filtered, washed with water (20 mL), and dried under reduced pressure.

[(5-Phenyl-1,3,4-oxadiazol-2-ylamino)(4-trifluoromethylphenyl)methyl]phos- phonic acid (**6a**). White solid, yield 92%. M.p. > 260 °C; ^1^H NMR (300 MHz, DMSO-d_6_): *δ* = 8.98 (br s, 1H, NH), 7.78–7.71 (m, 6H, arom. CH), 7.52–7.51 (m, 3H, arom. CH), 5.02 (d, 1H, CHP, ^2^*J*_PH_ = 21.7 Hz) ppm; ^13^C{^1^H} NMR (126 MHz, DMSO-d_6_): *δ* = 163.3 (d, arom. Cquat para of CF_3_, ^2^*J*_CP_ = 11.2 Hz), 158.1 (s, arom. Cquat C(Ph)=N), 142.3 (s, arom. Cquat C(NH)=N), 130.7 (s, arom. CH), 129.3 (s, arom. CH), 128.8 (d, arom. CH, ^3^*J*_CP_ = 4.0 Hz), 127.8 (q, arom. Cquat CCF_3_, ^2^*J*_CF_ = 32.3 Hz), 125.2 (s, arom. CH), 124.8 (s, arom. CH), 124.4 (q, CF_3_, ^1^*J*_CF_ = 272.7 Hz), 124.1 (s, arom. Cquat of C_6_H_5_), 56.0 (d, CHP, ^1^*J*_CP_ = 146.8 Hz) ppm; ^31^P{^1^H} NMR (121 MHz, DMSO-d_6_): *δ* = 14.3 (q, P(O), ^7^*J*_PF_ = 1.2 Hz) ppm; ^19^F{^1^H} NMR (282 MHz, DMSO-d_6_): *δ* = −60.8 (d, CF_3_, ^7^*J*_PF_ = 2.2 Hz) ppm. IR: *ν* = 1619 (C=N) cm^−1^. Elemental analysis calcd. (%) for C_16_H_13_O_4_N_3_F_3_P (399.26): C 48.13, H 3.28, N 16.03; found C 48.05, H 3.23, N 15.96.

[(5-Phenyl-1,3,4-oxadiazol-2-ylamino)(4-nitrophenyl)methyl]phosphonic acid (**6b**). White solid, yield 87%. ^1^H NMR (300 MHz, DMSO-d_6_): *δ* = 8.92 (s br, 1H, NH), 8.55 (s br, 2H, P(OH)_2_), 8.22 (d, 2H, arom. CH, ^3^*J*_HH_ = 8.7 Hz), 7.81–7.77 (m, 4H, arom. CH), 7.53–7.51 (m, 3H, arom. CH), 5.10 (d, 1H, CHP, ^2^*J*_PH_ = 22.5 Hz) ppm; ^13^C{^1^H} NMR (126 MHz, DMSO-d_6_): *δ* = 163.7 (d, arom. Cquat para of NO_2_, ^2^*J*_CP_ = 11.2 Hz), 158.6 (s, arom. Cquat C(Ph)=N), 147.2 (d, arom. Cquat C(NH)=N, ^3^*J*_CP_ = 2.1 Hz), 146.0 (s, arom. Cquat CNO_2_), 131.1 (s, arom. CH), 129.7 (s, arom. CH), 129.7 (d, arom. CH, ^3^*J*_CP_ = 3.9 Hz), 125.7 (s, arom. CH), 124.5 (s, arom. Cquat of C_6_H_5_), 123.5 (s, arom. CH), 56.6 (d, CHP, ^1^*J*_CP_ = 145.0 Hz) ppm; ^31^P{^1^H} NMR (121 MHz, DMSO-d_6_): *δ* = 13.8 (s, P(O)) ppm. IR: *ν* = 1623 (C=N) cm^−1^. Elemental analysis calcd. (%) for C_15_H_13_O_6_N_4_P (376.26): C 47.88, H 3.48, N 14.89; found C 47.76, H 3.61, N 14.75.

[(5-Phenyl-1,3,4-oxadiazol-2-ylamino)(4-fluorophenyl)methyl]phosphonic acid (**6c**). White solid, yield 95%. ^1^H NMR (300 MHz, DMSO-d_6_): *δ* = 8.73 (br s, 1H, NH), 7.82–7.79 (m, 2H, arom. CH), 7.57–7.51 (m, 5H, arom. CH), 7.17 (t, 2H, arom. CH, ^3^*J*_HH_ = 8.8 Hz), 4.94 (d, 1H, CHP, ^2^*J*_PH_ = 21.6 Hz) ppm; ^13^C{^1^H} NMR (126 MHz, DMSO-d_6_): *δ* = 163.3 (d, arom. Cquat of C_6_H_4_F, ^2^*J*_CP_ = 10.8 Hz), 161.5 (d, arom. Cquat CF, ^1^*J*_CF_ = 243.3 Hz), 158.0 (s, arom. Cquat C(Ph)=N), 133.5 (d, arom. Cquat C(NH)=N, ^3^*J*_CP_ = 2.3 Hz), 130.6 (s, arom. CH of C_6_H_5_), 130.5 (dd, arom. CH of C_6_H_4_F, ^3^*J*_CF_ = 7.6 Hz, ^3^*J*_CP_ = 5.3 Hz), 129.3 (s, arom. CH of C_6_H_5_), 125.2 (s, arom. CH of C_6_H_5_), 124.1 (s, arom. Cquat of C_6_H_5_), 114.7 (d, arom. CH of C_6_H_4_F, ^2^*J*_CF_ = 21.3 Hz), 55.4 (d, CHP, ^1^*J*_CP_ = 149.6 Hz) ppm; ^31^P{^1^H} NMR (121 MHz, DMSO-d_6_): *δ* = 15.2 (d, P(O), ^5^*J*_PF_ = 4.1 Hz) ppm; ^19^F{^1^H} NMR (282 MHz, DMSO-d_6_): *δ* = −115.6 (d, ^5^*J*_PF_ = 4.2 Hz) ppm. IR: *ν* = 1620 (C=N) cm^−1^. Elemental analysis calcd. (%) for C_15_H_13_O_4_N_3_FP (349.25): C 51.58, H 3.75, N 12.03; found C 51.45, H 3.67, N 11.96.

[(5-Phenyl-1,3,4-oxadiazol-2-ylamino)(3,4,5-trifluophenyl)methyl]phosphonic acid (**6d**). White solid, yield 89%. ^1^H NMR (300 MHz, DMSO-d_6_): *δ* = 9.91 (s br, 2H, P(OH)_2_), 8.74 (s br, 1H, NH), 7.81 (d, 2H, arom. CH, ^3^*J*_HH_ = 2.7 Hz), 7.52–7.45 (m, 5H, arom. CH), 5.00 (d, 1H, CHP, ^2^*J*_PH_ = 21.3 Hz) ppm; ^13^C{^1^H} NMR (126 MHz, DMSO-d_6_): *δ* = 163.1 (d, arom. Cquat of C_6_H_2_F_3_, ^2^*J*_CP_ = 10.7 Hz), 158.2 (s, arom. Cquat C(Ph)=N), 149.8 (dd, arom. Cquat CF, ^1^*J*_CF_ = 247.3 Hz, ^2^*J*_CF_ = 9.1 Hz), 137.9 (dt, arom. Cquat CF, ^1^*J*_CF_ = 250.5 Hz, ^2^*J*_CF_ = 14.9 Hz), 135.0 (d, arom. Cquat C(NH)=N, ^3^*J*_CP_ = 3.5 Hz), 130.7 (s, arom. CH), 129.3 (s, arom. CH), 125.3 (s, arom. CH), 124.1 (s, arom. Cquat of C_6_H_5_), 112.7 (d, arom. CH of C_6_H_2_F_3_, ^2^*J*_CF_ = 17.1 Hz), 55.2 (d, CHP, ^1^*J*_CP_ = 147.8 Hz) ppm; ^31^P{^1^H} NMR (121 MHz, DMSO-d_6_): *δ* = 13.98 (d, P(O), ^6^*J*_PF_ = 4.4 Hz) ppm; ^19^F{^1^H} NMR (282 MHz, DMSO-d_6_): *δ* = −135.9 (d, meta CF of C_6_H_2_F_3_, ^3^*J*_FF_ = 21.7 Hz), −163.5 (td, para CF of C_6_H_2_F_3_, ^3^*J*_FF_ = 21.7 Hz, ^6^*J*_PF_ = 4.4 Hz) ppm. IR: *ν* = 1612 (C=N) cm^−1^. Elemental analysis calcd. (%) for C_15_H_11_O_4_N_3_F_3_P (385.23): C 46.77, H 2.88, N 10.91; found C 46.69, H 2.95, N 10.87.

[(5-Phenyl-1,3,4-oxadiazol-2-ylamino)(3,5-bis-trifuoromethyl-phenyl)methyl] phosphonic acid (**6e**). White solid, yield 68%. ^1^H NMR (300 MHz, DMSO-d_6_): *δ* = 9.11 (s br, 1H, NH), 8.23 (s, 2H, arom. CH of C_6_H_3_(CF_3_)_2_), 8.01 (s, 1H, arom. CH of C_6_H_3_(CF_3_)_2_), 7.79 (d, 2H, arom. CH, ^3^*J*_HH_ = 8.7 Hz), 7.51–7.50 (m, 3H, arom. CH), 5.99 (s br, 2H, P(OH)_2_), 5.24 (d, 1H, CHP, ^2^*J*_PH_ = 21.9 Hz) ppm; ^13^C{^1^H} NMR (126 MHz, DMSO-d_6_): *δ* = 163.2 (d, arom. Cquat of C_6_H_3_(CF_3_)_2_, ^2^*J*_CP_ = 10.3 Hz), 158.2 (s, arom. Cquat C(Ph)=N), 141.5 (s, arom. Cquat C(NH)=N), 130.7 (s, arom. CH), 129.9 (q, arom. Cquat CCF_3_, ^2^*J*_CF_ = 32.6 Hz), 129.2 (s, arom. CH), 128.8 (s, arom. CH), 125.2 (s, arom. CH), 124.0 (s, arom. Cquat of C_6_H_5_), 123.5 (q, CF_3_, ^1^*J*_CF_ = 273.3 Hz), 121.0 (s, arom. CH), 55.7 (d, CHP, ^1^*J*_CP_ = 144.6 Hz) ppm; ^31^P{^1^H} NMR (121 MHz, DMSO-d_6_): *δ* = 13.5 (s, P(O)) ppm; ^19^F{^1^H} NMR (282 MHz, DMSO-d_6_): *δ* = −61.2 (s, CF_3_) ppm. IR: *ν* = 1622 (C=N) cm^−1^. Elemental analysis calcd. (%) for C_17_H_12_O_4_N_3_F_6_P (467.26): C 43.70, H 2.60, N 8.99; found C 43.58, H 2.74, N 8.92.

### 2.5. X-ray Crystal Structure Analysis

Single crystals of **5b** suitable for X-ray analysis were obtained by slow diffusion of hexane into a dichloromethane solution of the *α*-aminophosphonate. Crystal data: C_19_H_21_N_4_O_6_P, *M*_r_ = 432.37 g mol^−1^, monoclinic, space group *P*2_1_/c, *a* = 18.472(2) Å, *b* = 5.257(2) Å, *c* = 21.00(2) Å, *β* = 91.268(18)°, *V* = 2039(4) Å^3^, *Z* = 4, *D* = 1.409 g cm^−3^, *μ* = 0.180 mm^−1^, *F*(000) = 904, *T* = 120(2) K. The sample (0.220 × 0.100 × 0.100) was studied on a Bruker PHOTON-III CPAD using Mo-*K*_α_ radiation (λ = 0.71073 Å). The data collection (2*θ*_max_ = 31.837°, omega scan frames by using 0.7° omega rotation and 30 s per frame, range *hkl*: *h* −27,27 *k* −6,6 *l* −24,24) gave 37,576 reflections. The structure was solved with SHELXT-2014/5 [39], which revealed the non-hydrogen atoms of the molecule. After anisotropic refinement, all of the hydrogen atoms were found with a Fourier difference map. The structure was refined with SHELXL-2018/3 [40] by the full-matrix least-square techniques (use of *F* square magnitude; *x*, *y*, *z*, *β*_ij_ for C, N, O and P atoms; *x*, *y*, *z* in riding mode for H atoms); 261 variables and 3554 observations with *I* > 2.0 *σ*(*I*); calcd. *w* = 1/[*σ*^2^(*F*o^2^) + (0.0946*P*)^2^ + 3.4472*P*] where *P* = (*F*o^2^ + 2*F*c^2^)/3, with the resulting *R* = 0.0763, *R*_W_ = 0.2188 and *S*_W_ = 1.091, Δ*p* < 0.915 eÅ^−3^. CCDC entry 2098845 contains the supplementary crystallographic data for **5b**. These data can be obtained free of charge from The Cambridge Crystallographic Data Centre via http://www.ccdc.cam.ac.uk/data_request/cif (accessed on 23 July 2021), by emailing data_request@ccdc.cam.ac.uk, or by contacting The Cambridge Crystallographic Data Centre, 12 Union Road, Cambridge CB2 1EZ, UK

### 2.6. Biological Activities

#### 2.6.1. Cells and Virus

Vero cells (CCL-81, American Type Culture Collection, Boston, MA, USA), used as a cellular model for antiviral bioassay, were cultured in Dulbecco’s modified Eagle’s medium (DMEM), supplemented with 10% fetal bovine serum (FBS) and gentamycin, and incubated at 37 °C with 5% CO_2_. They were used to adapt the avian Massachusetts serotype CoV-IBV strain.

#### 2.6.2. Cell Toxicity and Viability

Cell viability was evaluated using the MTT (3-(4,5-dimethylthiazol-2-yl)-2,5-diphe- nyltetrazolium bromide) method [41]. The MTT assay is sensitive and reliable for cellular metabolic activity. It is based on the ability of nicotinamide adenine dinucleotide phosphate (NADPH)-dependent cellular oxidoreductase enzyme to reduce the tetrazolium dye to the purple formazan crystals. Sub-confluent Vero cells, seeded in 96-well microplates (2 × 10^6^ cells/microplate; Thermo Fisher Scientific), were incubated with serial dilutions of synthesized molecules, in duplicates for 3 days at 37 °C. MTT solution (500 μg/mL) was then added, and the cells were incubated for further 4 h. The formed formazan crystals within metabolically viable cells were then dissolved in 100 µL of DMSO, and the absorption, proportional to viable cell number, was measured at 490 nm with a Thermo Scientific Multiskan FC. The results were expressed as a percentage of viable cells relative to the negative control (untreated cells cultured in standard medium and considered 100% viable). The half-maximum cytotoxic concentration (CC_50_), which was defined as the concentration which reduced the OD_490_ of the 1,3,4-oxadiazole-derived *α*-aminophosphonates/*α*-aminophosphonic acids-treated cells to 50% of that of untreated cells, were determined from the plotted curve of percentages of viable cells against compound concentrations. 

#### 2.6.3. Virucidal Assay

Avian CoV-IBV (10^4^ TCID_50_; 50% tissue-culture infectious dose of Avian CoV-IBV) was incubated with α-aminophosphonates/α-aminophosphonic acids in 200 µL of DMEM containing 5% of FBS for 2 h at 37 °C with 5% CO_2_. The suspensions were then added to confluent Vero cells in 24-well microplates (2 × 10^6^ cells/microplate; Thermo Fisher Scientific), and the mixture was incubated for 2 h at 37 °C with 5% CO_2_. The cell layers were washed with phosphate-buffered saline (PBS; 500 µL). DMEM containing 5% FBS (500 µL) was added, and the cells were incubated for 3 days at 37 °C with 5% CO_2_. Cells and supernatants were harvested, freeze-dried, and then centrifuged before the determination of virus content. The residual viral infectivity was determined by qRT-PCR, and the percentage of inhibition was deduced. 

#### 2.6.4. Avian CoV-IBV Replication Inhibition Assay

The effect of α-aminophosphonates/α-aminophosphonic acids on avian CoV-IBV infection was evaluated using virus RNA quantification. Vero cells at 2 × 10^6^ cells/plate were seeded in 24-well microplates, inoculated with 10^4^ TCID_50_ (50% tissue–culture infectious dose of Avian CoV-IBV), and incubated for 2 h. The cell supernatants were then removed, and the layers washed twice with PBS, (pH 7.2); then, serial dilutions of the tested compounds were added to the infected cells. Wells containing Vero cells culture medium were considered as a negative control. After incubation for 72 h, culture cells and supernatants were harvested, freeze-dried 3 times, and cleared by centrifugation at 500× *g* for 15 min at 4 °C. The collected supernatants were then labeled and stored at −80 °C until titration.

#### 2.6.5. Quantitative Real-Time PCR (qRT-PCR)

Avian CoV-IBV genomic RNA was extracted from the supernatant of infected Vero cells after cell lysis, at 3 days post-infection, using the IndiSpin^®^ Pathogen Kit (Indical Bioscience GmbH). The extracted nucleic acids were resuspended in the elution buffer (100 µL), and RNase inhibitor (RiboGrip RNase Inhibitor, Solis BioDyne) was added (0.4 µL) and stored at −80 °C until use.

The quantitative Real-Time PCR (qRT-PCR) was carried out using LightCycler 2.0 (Roche Diagnostics). For detection of genomic avian CoV-IBV, a forward primer IBV5′GU391 (5′-GCT TTT GAG CCT AGC GTT-3′), located at nucleotide positions 391–408, a reverse primer IBV5′GL533 (5′-GCC ATG TTG TCA CTG TCT ATT G-3′), located at nucleotide positions 533–512, and a Taqman^®^ dual-labeled IBV5′G probe (5′-FAM-CAC CAC CAG AAC CTG TCA CCT C-BHQ1-3′), located at nucleotide positions 494–473, were used [42]. The qRT-PCR steps comprised 50 °C for 15 min, 95 °C for 10 min, and 40 cycles of 95 °C for 15 s and 60 °C for 1 min.

The primers and probes contained 5× One-step Probe Mix (4 µL), 40× One-step SOLIScript^®^ Mix (0.5 µL), and probe (0.5 µL). The final volume was adjusted with nuclease-free water to obtain 20 µL with a concentration of 0.2 µM. Following the manufacturer’s instructions, a standard curve was determined for the assay and used to calculate genomic copy numbers for each sample. The standard curve was generated using 10-fold dilutions of positive control.

The amplification efficiency was calculated using the LightCycler 480 software (Roche Lifescience). The qRT-PCR efficiency was deduced from the standard curve. Briefly, a positive standard curve was generated by performing qRT-PCR on a series of diluted avian CoV-IBV templates, ranging from 10^7^ to 10^1^ 50% tissue-culture infectious dose (TCID_50_) per microliter. The PCR efficiency was calculated using the equation E = 10[−1/Slope] × 100 [43]. The qRT-PCR software would provide a standard curve and a slope by measuring the fluorogenic quantification cycles, which are represented as cycle threshold (Ct) values versus Log10 virus gene copies (see the Appendix A). The resulting standard curve allowed the deduction of virus concentrations in cultures of Vero cells after each antiviral activity assay.

#### 2.6.6. Statistical Analysis

All data were normally distributed and presented as the mean ± standard deviation of triplicate. Comparison between means was performed using one-way analysis of variance (ANOVA) followed by Bonferonni’s post hoc test at the significance level *p* < 0.05.

### 2.7. Molecular Docking Method

DFT-based calculations of molecules **5f** and **6a** were carried out using ORCA version 4.1 before performing molecular docking analysis [44]. The x-ray crystal structure of **5b** was evaluated as a raw structure for performing the optimization procedure. The calculations were carried out by using the exchange function according to BP86, as suggested by Becke and Perdew [45,46]. Tightscf and grid4 options and resolution-of-the-identity (RI) approximation were used with the TZV basis set, and the TZV/J auxiliary basis set was used for speeding up the performing calculations [47]. Crystal structure of SARS coronavirus main proteinase (PDB ID: 1uk4) [48], papain-like protease (PDB ID: 4x2z) [49], and N- and C-terminal domain of coronavirus nucleocapsid protein (PDB ID: 2c86) [50] were downloaded from RCSB protein data bank (https://www.rcsb.org/ (8 November 2022)). AutoDockTools 4.2 was used for molecular docking calculations. Only polar hydrogen and Kollman charges were evaluated in target molecules, and molecules of water inside proteins were removed. While Lamarckian genetic algorithms were applied, the genetic algorithm population was recorded as 150. Randomized starting positions, Gasteiger charges, and torsions have been evaluated for molecules **5f** and **6a** [51,52]. Discovery Studio 4.1.0 were used for illustrations.

## 3. Results and Discussion

### 3.1. Synthesis of α-Aminophosphonates and α-Aminophosphonic Acids

It is well known that the Kabachnik–Fields reaction [53,54], which involves the one-pot three-component condensation of amines, aldehydes, and dialkyl phosphites in the presence of acid catalysts, is undoubtedly the most straightforward approach to form *α*-aminophosphonates [55]. First, we applied this classical method to the target *α*-aminophosphonates from 5-phenyl-1,3,4-oxadiazol-2-amine (**1**) in the presence of benzaldehyde and diethyl phosphite with various acid catalysts such as ZnCl_2_, FeCl_3,_ or AcOH with or without solvent [56,57]. Unfortunately, under these conditions, the desired *α*-aminophosphonate could only be isolated in very low yields (<5%). Furthermore, carrying out the reaction under microwave irradiation did not lead to the desired product but to the exclusive formation of an *α*-hydroxyphosphonate resulting from the condensation between the benzaldehyde and the diethyl phosphite. This latter result suggests that the formation of the expected imine intermediate did not occur under these operating conditions.

To circumvent this problem, the Pudovik reaction [58], which is the stepwise version of the Kabachnik-Fields condensation, requiring the isolation of the imine intermediate, was attempted. Thus, imine intermediates **2a**–**f** were first prepared and isolated from the reaction of 5-phenyl-1,3,4-oxadiazol-2-amine (**1**) with aromatic aldehydes (1.5 equiv.) in refluxing toluene for 48 h (Figure 1). Imines **2a**–**f** were isolated as solids in good to excellent yields (73–90%) and fully characterized by elemental analysis, infrared, and NMR spectroscopies (^1^H, ^13^C, and ^19^F); ^1^H NMR spectra, in particular, displayed a signal around 9.34–9.58 ppm attributable to the N=CH proton (see Appendix A).

With imines **2a**–**f** in hand, we then studied their reactivity in the presence of dialkyl/arylphosphites, in order to access the target *α*-aminophosphonates. For the Pudovik reaction, we chose to use an environmentally friendly protocol. The condensation was realized under microwave irradiation in solvent- and catalyst-free conditions, which have become a modern approach for the synthesis of bioactive heterocyclic molecules [59,60,61]. The optimization was carried out with imine **2a** and dimethyl phosphite. After 10 min at 80 °C and 60 W, the expected α-aminophosphonate **3** to be isolated with a moderate yield of 20% (Table 1, entry 1). A longer irradiation time did not increase the yield of the desired product (Table 1, entry 2). A significantly improved isolated yield of 54% was obtained when the reaction was conducted at a higher temperature (115 °C) with a higher power (300 W) for only 10 min (Table 1, entry 3). It is important to note that under these operating conditions and using either diphenyl phosphite or diethyl phosphite, the reaction provided the corresponding *α*-aminophosphonates **4** and **5a** with isolated yields of 90 and 92%, respectively (Table 1, entries 4 and 5).

Under optimized conditions (Table 1, entry 5), the six imines **2a**–**f** bearing either electron-donating or electron-withdrawing substituents on the phenyl ring, were put in the presence of diethyl phosphite to generate the corresponding 1,3,4-oxadiazole-derived *α*-aminophosphonates **5a**–**f** (Figure 2). It is interesting to note that imines with electron-withdrawing substituents (**2a**, **2b**, and **2e**) led to corresponding *α*-aminophosphonates (**5a**, **5b**, and **5e**) in lower isolated yields (76–92%) than substrates bearing electron-donating substituents, mesomeric effect (**2c**, **2d**, and **2f**; yields 90–97%). The difference in reactivity could be attributed to the substituent’s effects, such as strong electron-donating groups on the phenyl ring, which would increase the electron density on the imine nitrogen atom and, therefore, enhance the electrophile of the carbon atom, which favors the hydrophosphonylation reaction, as suggested by the mechanism reported earlier by the Cherkasov’s group [62].

The synthesized 1,3,4-oxadiazole-derived *α*-aminophosphonates **3**, **4**, and **5a**–**f** were fully characterized by elemental analysis, infrared and multinuclear NMR spectroscopies (see the experimental part). Their ^1^H NMR spectra in DMSO-d_6_ revealed for each NH signal a doublet of a doublet in the range 9.00–9.55 ppm (*^3^J*_HH_ = 6.0–10.2 Hz and *^3^J*_PH_ = 1.5–3.9 Hz and) and for each CHP signal a doublet of a doublet in the range 5.24–5.95 ppm (*^2^J*_PH_ = 13.2–23.1 Hz and *^3^J*_HH_ = 6.0–10.2 Hz). Their ^31^P{^1^H} NMR spectra show a signal centered in the attempt range of 12.6–21.7 ppm (see Appendix A). 

Furthermore, the structure of *α*-aminophosphonates was confirmed by an X-ray diffraction study carried out on compound **5b** (Figure 2). The α-aminophosphonate crystallizes in the monoclinic asymmetric space group *P*2_1_/c with four distinct enantiomeric molecules (A and B molecules in which C1 has an R and an S configuration). The oxadiazole and phenyl aromatic rings are slightly inclined with a dihedral angle of 12.86°. The aromatic ring bearing the nitro substituent is perpendicular to the oxadiazole ring; the dihedral angle between these two aromatic rings is 88.73° (Figure 2). Bond lengths and angles of the 1,3,4-oxadiazole ring are consistent with those of **5a**, previously reported by our group [37] (see Appendix A).

*α*-Aminophosphonates **5a**–**e** were then hydrolyzed into their corresponding *α*-aminophosphonic acids **6a**–**e**. The dealkylation was carried out in mild conditions using eight equivalents of trimethylsilyl bromide (TMSBr) at room temperature for three days. After acid hydrolysis, the *α*-aminophosphonic acids **6a**–**e** were isolated in good to excellent yields and purities (68–95%; Figure 3). Compounds **6a**–**e** were fully characterized by elemental analysis, infrared and multinuclear NMR spectroscopies (see experimental section and Appendix A). Their ^1^H NMR spectra showed, in particular, the disappearance of the ethoxyl groups, while their ^31^P{^1^H} NMR spectra displayed a downfield of about 5 ppm.

### 3.2. Biological Activity

A selection of synthesized α-aminophosphonates **5b**,**c**,**f**, and *α*-aminophosphonic acids **6a**–**e** were screened for their antiviral activity against avian coronavirus IBV.

#### 3.2.1. Determination of Cell Growth and Viability

The MTT-based (3-(4,5-dimethylthiazol-2-yl)-2,5-diphenyltetrazolium bromide) cytotoxicity assay [41] was used to determine the safety profiles of the test compounds on Vero cells. As shown in Figure 3, the *α*-aminophosphonate **5c** and the *α*-aminophosphonic acids **6a**, **6b**, **6d**, and **6e** were the highest cytotoxic compounds with CC_50_ values of around 550 µM. The diethyl[(5-phenyl-1,3,4-oxadiazol-2-ylamino) (4-nitrophenyl)methyl]phosphonate (**5b**) was the lowest cytotoxic compound having a CC_50_ value of 1598.8 ± 4.73 µM. The other 1,3,4-oxadiazole-derivatives showed intermediates CC_50_ values (>660 µM). The cytotoxic profile of the test compounds was established as follows: **6d** ≥ **6b** > **5c** > **6a** > **5f** > **6e** > **6c** > **5b**. The important CC_50_ values (≥540 µM) encourage their use as safe and non-cytotoxic bioactive compounds.

The safety of *α*-aminophosphonates has already been reported for a series of ribonucleosides of 1,2,3-triazolylbenzyl-aminophosphonates, which have minimum cytotoxic concentrations of higher than 100 µM in Vero cell cultures [63]. Rezaei’s group observed a moderate cytotoxicity profile of some α-aminophosphonates (>100 µM minimum cytotoxic concentration) against the three cancer cell lines, including JURKAT (T-cell lymphoma), RAJI (Burkit’s lymphoma), and MCF-7 (breast cancer) [64]. In contrast, a strong inhibitory effect of a family of thiazolyl *α*-aminophosphonates against five human cancer cell lines, including breast (MCF-7 and MDA-MB-231), prostate (DU-145), liver (HepG2), and HeLa cancer cell lines, has been reported by Cirandur’s group [65].

#### 3.2.2. Determination of Cell Growth and Viability

The inhibitory effect of compounds **5b**,**c**,**f**, and **6a**–**e** on avian CoV-IBV was first tested using the direct assay, which consists of mixing a known amount of virus with aliquots containing the organic molecule. In this aim, the inhibition of the replication of the virus in Vero cells was measured. Data from Table 2 showed that the direct contact of test compounds with virus particles resulted in 22.73–86.11% inhibition of viral replication. The *α*-aminophosphonate **5f** (at 33 µM) and the α-aminophosphonic acid **6a** (at 1.23 µM) proved to be by far the most effective compounds exhibiting 86.11 ± 1.58 and 75.00 ± 0.75% inhibition of IBV, respectively. This indicates that both compounds have virucidal activities as they are able to inactivate the IBV infectivity outside cells. 

To further evaluate whether the compounds **5b**,**c**,**f**, and **6a**–**e** have antiviral activity, the IBV genomic RNA was extracted from infected Vero cells after three days post-infection, and the virus titer was quantified using a quantitative Real-Time PCR (qRT-PCR). As reported in Table 2, all tested compounds failed to inhibit the replication of the IBV virus into the host cells, which demonstrates their ineffectiveness as antiviral agents. At this stage, the inability of the test compounds to penetrate the host cell due to their polar character may explain these negative results. These results were consistent with those observed by Lazrek’s group, which tested ten nucleosides of 1,2,3-triazolylbenzyl-aminophosphonates against a broad range of DNA and RNA viruses in Vero cell cultures and measured the absence or a very weak anti-viral activity of the test products [63]. Similarly, Song’s group obtained low antiviral activity against the Tobacco Mosaic Virus using *α*-aminophosphonates. Moreover, the activities drastically depend on the nature of the substituents of the nitrogen and phosphorus atoms [66].

Compared to lithium chloride (LiCl), a potent inhibitor of the DNA of the virus, the tested *α*-aminophosphonates **5b**,**c**,**f**, and *α*-aminophosphonic acids **6a**–**e** were found to be far more effective virucidal agents against IBV at the concentration range used. Thus, it has been shown that incubation of LiCl at concentrations ranging from 5 to 50 mM with the IBV Beaudette strain did not modify the title of the virus, indicating its inability to generate virucidal activity against IBV. Furthermore, LiCl had the ability to suppress the replication of IBV in Vero cells in a dose-dependent manner without a virucidal effect. The authors attributed the observed activity to the inhibitory effect of LiCl on host cell protein, such as glycogen synthase kinase-3, thus suppressing the IBV nucleocapsid (N) protein [12]. In other comparative studies, ethanolic extracts from 15 plant species were used. High virucidal activities against the IBV Beaudette strain were observed for extracts mainly derived from the Lamiaceae family, including Satureja Montana, Origanum vulgare, Mentha piperita, Melissa officinalis, Thymus vulgaris, Hyssopus officinalis, Salvia officinalis, and Desmodium canadense with 50% effective concentrations (EC_50_) in the microgram range (0.003–0.076 µg) with strong inhibitory effects on IBV replication in Vero cells at concentrations of 0.27–0.63 µg [10]. More recently, a formulation made from plant essential oils of cinnamaldehyde and glycerol monolaurate (5:95), in vivo, was found to have great potential as an anti-IBV agent. In their animal model (Yellow feather broiler chicks), Mo’s group demonstrated that the mixture of essential oils inhibits virus replication, promotes immune function, and reduces the release of the pro-inflammatory cytokine interleukin-6 [11].

#### 3.2.3. Molecular Docking

The recent developments in computational chemistry have strengthened the agreement between experimental and computed results [67]. Theoretical methods, which initially focused on structural analysis of the molecules, are also currently used in the evaluation of intermolecular interactions. The modern methods, such as molecular docking that examine the interactions between biomacromolecules and drug candidates, are an important part of structure-based drug design [68,69]. In the aim to obtain information on the mode of action, we studied the interactions between *α*-aminophosphonate **5f** and *α*-aminophosphonic acid **6a**, which have the highest virucidal activities against avian CoV-IBV; SARS coronavirus main proteinase, papain-like protease, and N- and C-terminal domain of coronavirus nucleocapsid protein were analyzed.

First, the most stable structures of **5f** and **6a** were optimized with the ORCA package [44]. Both compounds were recorded in pdbqt format. For the molecular docking study, the macromolecules were downloaded from the CBSN protein database, then blinded docking was performed to determine initial interactions, and the detected regions were examined in detail. Exposures with the lowest standard deviation and binding constant were recorded.

The coronavirus polyprotein encodes two proteases, the main protease and the papain-like protease [70]. These two enzymes have been frequently chosen as target molecules in drug development studies related to the COVID-19 pandemic [71]. Although both experimental and theoretical analyses have been performed for many types of molecules during the COVID-19 pandemic, to the best of our knowledge, *α*-aminophosphonates and *α*-aminophosphonic acids have not been studied by molecular docking for interactions with coronavirus main protease and papain-like protease. Nevertheless, molecular docking interactions of phosphonimidates containing chitosan moieties against protease of SARS coronavirus were analyzed and displayed remarkable binding affinity (−7.9 to −7.3 kcal/mol) [72]. On the other hand, in 2006, Wu’s group investigated the action of 58,855 compounds on inhibition of the main protease of the coronavirus and detected, through a virtual structure-based analysis, two types of effective compounds, one of which had an oxadiazole moiety [73]. Therefore, interactions of *α*-aminophosphonate **5f** and *α*-aminophosphonic acid **6a** with the main protease enzyme of the coronavirus were first investigated. The two drugs interacted with the same region of the crystalline structure; the interaction of the phosphonic acid **6a** is more important than with the phosphonate **5f**. An inhibition concentration of 16.4 µM corresponding to a binding affinity of −6.53 kcal/mol was calculated for **6a**, while a binding affinity of −5.98 kcal/mol and an inhibition constant of 41.28 µM were evaluated for **5f**. In addition to the van der Waals interactions, H-bonds between Gln110, Thr111, Asn151, Thr292, and the phosphonic acid region, and effective π-interactions between Leu202 and Phe294 and the conjugated cyclic regions of the molecule were detected for **6a**. H-bonds with Ile106, Thr111, and Asn151 were observed for **5f**. Other notable interactions for the latter drug are the halogenic interactions between Arg105, Gln110, and the CF_3_ moiety. These interactions are represented in Figure 4.

Slightly higher interactions for **6a** (binding affinity of −5.85 kcal/mol and inhibition constant of 43.36 µM) than **5f** (binding affinity of −5.08 kcal/mol and inhibition constant of 187.73 µM) were calculated with papain-like protease. In the case of **6a**, H-bonds were observed with Asn155 and Ala237, while with **5f**, only H-bonds with Ser152 were recorded. The observed halogenic interactions for **5f** and **6a** played a positive role in the formation of the inclusion complexes (see Appendix A).

Enveloped viruses, such as coronavirus, have a filamentous nucleocapsid structure formed by the interaction between nucleocapsid (N) protein and single-stranded viral RNA [74]. The nucleocapsid protein plays an important role in viral genome replication and cell signaling pathways [75]. Therefore, in this study, we also examined the interactions of **5f** and **6a** with the N- and C-terminal domains of coronavirus nucleocapsid protein. In the case of *α*-aminophosphonate **5f**, a binding affinity of −5.01 kcal/mol and an inhibition constant of 214.16 µM were calculated; the highest contribution to these values comes from H-bonds with Ser29, Asn32, Ser34, Gln74, Tyr94, and Arg132. For the *α*-aminophosphonic acid **6a**, the measured interactions with the protein were more important (binding affinity of −6.56 kcal/mol and inhibition constant of 15.6 µM) and mainly halogenic interaction with Met28 and H-bonds with Ser29, Asn32, Ser34, Arg72, and Tyr94 were observed (see Appendix A).

## 4. Conclusions

In summary, we have described the synthesis of a family of dialkyl/aryl[(5-phenyl- 1,3,4-oxadiazol-2-ylamino)(aryl)methyl]phosphonates through the Pudovik-type reaction of dialkyl/arylphosphite with imines under microwave irradiation and their hydrolyzed phosphonic acids derivatives. A selection of non-cytotoxic compounds was screened for their in vitro antiviral activity against the avian bronchitis virus and displayed moderate virucidal inhibition. The more efficient 1,3,4-oxadiazole-based drugs, *α*-aminophosphonate **5f** (at 33 µM) and *α*-aminophosphonic acid **6a** (at 1.23 µM), could be useful for the prevention of IBV infection with virucidal inhibition of 86.11 ± 1.58 and 75.00 ± 0.75%, respectively, allowing additional future studies, their incorporation into cleaning products, disinfectants, wipes, and nasal sprays. In order to better understand the mechanism of action of **5f** and **6a**, their interactions with the main protease, the papain-like protease, and the nucleocapsid protein were investigated by molecular docking. Remarkable binding and inhibition constants were recorded for these two compounds. However, the tested compounds were unable to inhibit virus replication in IBV-infected Vero cells. The lack of antiviral activity could probably be attributed to their polarity, which hinders their diffusion through the lipophilic cytoplasmic membrane. To circumvent this problem and promote their diffusion into the host cell and, thus, their antiviral efficacy, chemical modification such as monoesterification of *α*-aminophosphonic acids [76] or encapsulation of the most promising compounds in liposomes/lipid-based carriers/nanocarriers [77,78,79,80] will be the subject of future investigations.

## Data Availability

Not applicable.

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
