# Peer review of "Synthesis of Novel 1,3,4-Oxadiazole-Derived α-Aminophosphonates/α-Aminophosphonic Acids and Evaluation of Their In Vitro Antiviral Activity against the Avian Coronavirus Infectious Bronchitis Virus"

_pharmaceutics, 2022, doi:10.3390/pharmaceutics15010114_

Round 1

Reviewer 1 Report

S. Hkiria and co-workers submitted the manuscript intitled: Synthesis of novel 1,3,4-oxadiazole-derived α-aminophosphonates/α-aminophosphonic acids and evaluation of their in vitro antiviral activity against the avian coronavirus infectious bronchitis virus.

α-Aminophosphonates, which are structural analogues of natural α-amino acids, have elicited considerable interest from chemists due to their wide-range of useful properties. In addition to their well-known biological activities, these phosphorus compounds have gradually developed from a synthetic challenge to a directed and rational design of novel molecular ligands with outstanding metal-complexing properties. Oxadiazole derivatives, especially the 1,3,4-oxadiazole isomer, have wide pharmacological applications and coordination properties that are greatly described in introduction part of the manuscript. For these reasons, the synthesis of new 1,3,4-oxadiazole-derived α-aminophosphonates/α-aminophosphonic acids represent a good novelty in the pharmacological  scientific research.

Manuscript is very well structured in all parts, the introduction is clear and describes very well the authors' intent in conducting the experimental work. The experimental part was conducted with excellent scientific rigor and is well supported by a rich supporting information, in fact, the spectral characterization is very complete and exaustive. Synthetic part is exhaustive and biological data shown good results because the used concentration of bioactive compounds is in the order of μM.

Based on the above, I strongly recommend the publication of the manuscript in Pharmaceutics.

In according to the authors, I suggest in row 439 the introduction of a sentence that emphasizes the use of microwaves as a modern approach for the synthesis of numerous bioactive heterocyclic molecules (ref.1,2).

11) A. Adhikari, S. Bhakta, T. Ghosh. Tetrahedron, 2022, 126, 133085.2

22) L. Maiuolo, V. Algieri, B. Russo, M. A. Tallarida, M. Nardi, M. L. Di Gioia, Z. Merchant, P. Merino, I. Delso, A. De Nino. Molecules, 2019, 24, 2909.

33) R. N. Rao, S. Jena, M. Mukherjee, B. Maiti, K. Chanda. Environmental Chemistry Letters, 2021, 19, 3315–3358.

Best Regards

Author Response

We thank this referee for his/her constructive comments

In according to the authors, I suggest in row 439 the introduction of a sentence that emphasizes the use of microwaves as a modern approach for the synthesis of numerous bioactive heterocyclic molecules (ref.1,2).

1) A. Adhikari, S. Bhakta, T. Ghosh. Tetrahedron, 2022, 126, 133085.2

2) L. Maiuolo, V. Algieri, B. Russo, M. A. Tallarida, M. Nardi, M. L. Di Gioia, Z. Merchant, P. Merino, I. Delso, A. De Nino. Molecules, 2019, 24, 2909.

3) R. N. Rao, S. Jena, M. Mukherjee, B. Maiti, K. Chanda. Environmental Chemistry Letters, 2021, 19, 3315–3358.

The following comment “which have become a modern approach for the synthesis of bioactive heterocyclic molecules [60-62]” and the three references were added.

Reviewer 2 Report

The manuscript titled Synthesis of novel 1,3,4-oxadiazole-derived α-aminophosphonates/α-aminophosphonic acids and evaluation of their in vitro antiviral activity against the avian coronavirus infectious bronchitis virus by authors et al. demonstrated that the direct contact of novel synthesized compounds with IBV showed that the diethyl[(5-phenyl-1,3,4-oxadiazol-2-ylamino)(4-trifluoromethoxyphenyl) methyl]phosphonate (at 33 µM) and the [(5-phenyl-1,3,4-oxadiazol-2-ylamino)(4-trifluoromethylphenyl)methyl]phosphonic acid (at 1.23 µM) strongly inhibited the IBV infectivity. In general, this paper seems to be quite interesting and I would like to recommend the acceptance of this work provided that authors can well address the following questions.

1. A brief conformational analysis is recommended, and molecular docking between the compound and the viral protein could also be done to improve the quality of the article.

2. How do these compounds compare with commercially available topical disinfectants?

Author Response

We thank the referee for his/her insightful remarks. 

A brief conformational analysis is recommended, and molecular docking between the compound and the viral protein could also be done to improve the quality of the article.”

Molecular docking calculations between compounds 5f and 6a and coronavirus main protease, papain-like protease and nucleocapsid protein were carried out by Elvan Üstün. Her name was added to the authors.

How do these compounds compare with commercially available topical disinfectants?

Although it is difficult to compare the virucidal activity of 1,3,4-oxadiazole-based α-aminophosphonate 5f and α-aminophosphonic acid 6a with the commercially available topical disinfectants due to differences in the technical conditions (concentration, mode of application, virus strains, etc.), it can be easily inferred that commercial preparation based on ethanol, sodium hypochlorite, sodium hydroxide, oximacro, hypochlorus acid water, argovite, Argovit, Triviron, Ecocid, lauric acid monoglyceride etc. were by far most active than 5f (at 33 µM) and 6a (at 1.23 µM). However, they could be used as adjuvants to empower the virucidal activity against IBV.

Reviewer 3 Report

The contents of manuscript is interesting dealing with synthesis and biological activity of selected 1,3,4-oxadiazolo-3alpha-aminophosphonates. These compounds were screened for their in vitro antiviral activity against the avian bronchitis virus (IBV). The bases for selection of target compounds structures are not clear to readers.  The chemical synthesis was based on classical Kabachnik-Fields reaction. This reaction is widely used for synthesis bioactive molecules  (see Materials 15 3846). Unfortunately direct three component reaction failed and therefore two step approach was developed. In experimental part the accuracy of NMR method for carbon and fluoride nucleus is only one digit after comma. The authors routinely used elemental analysis for structure assignments what should be nowadays highly appreciated. Biological data require special attention. For evaluation of obtained data statistical analysis should be used. The accuracy of virucidal activity inhibition for compounds 5 and 6 (Table 2) is definitely over 0.01%!! All compounds were not active against Avian infectious bronchitis virus. Two compounds 5f and α-aminophosphonic acid 6a could be useful for the prevention of IBV infection with moderate virucidal inhibition of 86.1 and 75.0 %, respectively. Their potential application in cleaning products or nasal sprays, requires additional studies.

The conclusion that the lack of antiviral activity could be attributed to their polarity, which hinders their diffusion through the lipophilic cytoplasmic membrane is quite interesting. Polarity of aminophosphonic acid can be modulated by selective monoesterification protocol (see Molecules 2021, 26, 5637). Products of this reaction 6 of monoester structure are known to efficiently  diffuse into the cells what may give the authors compounds with expected antiviral efficacy.

Author Response

We thank this referee for his/her constructive comments

The chemical synthesis was based on classical Kabachnik-Fields reaction. This reaction is widely used for synthesis bioactive molecules (see Materials 15 3846).

The reference was added (number 56).

In experimental part the accuracy of NMR method for carbon and fluoride nucleus is only one digit after comma.”

Modification was made.

For evaluation of obtained data statistical analysis should be used.”

This was made in caption of Figure 3.

The accuracy of virucidal activity inhibition for compounds 5 and 6 (Table 2) is definitely over 0.01%!! All compounds were not active against Avian infectious bronchitis virus. Two compounds 5f and α-aminophosphonic acid 6a could be useful for the prevention of IBV infection with moderate virucidal inhibition of 86.1 and 75.0 %, respectively.

The reviewer was right on this point and we fully agree with him. However, we would like to clarify that, despite the precision is greater than 0.01%, it remains in the acceptable range as indicated in many articles dealing with virucidal activity such as:

  • V. Torres, M. J. Domínguez, J. L. Carbonari, M. C. Sabini, L. I. Sabini, S. M. Zanon, Study of antiviral and virucidal activities of aqueous extract of Baccharis articulate against Herpes sui virus. Nat. Prod. Commun. 2011, 6, 993-994.
  • M. Bonotto, F. Bonì, M. Milani, A. Chaves-Sanjuan, S. Franze, F. Selmin, T. Felicetti, M. Bolognesi, S. Konstantinidou, M. Poggianella, C. L. Márquez, F. Dattola, M. Zoppè, G. Manfroni, E. Mastrangelo, A. Marcello, Virucidal activity of a pyridobenzothiazolone derivative HeE1-17Y against enveloped RNA viruses. Viruses, 2022, 14, 1157.
  • H. C. Lagrota, M. D. Wigg, M. M. G. Santos, M. M. F. S. Miranda, F. P. Camara, J. N. S. S. Couceiro, S. S. Costa, Inhibitory activity of extracts of Alternantherabrasiliana (Amaranthaceae) against the Herpes simplex visrus. Phytother. Res. 1994, 8, 358-361.
  • E. Siniavin, M. A. Streltsova, M. A. Nikiforova, D. S. Kudryavtsev, S. D. Grinkina, V. A. Gushchin, V. A. Mozhaeva, V. G. Starkov, A. V. Osipov, S. C. R. Lummis, V. I. Tsetlin, Y. N. Utkin, Snake venom phospholipase A2s exhibit strong virucidal activity against SARS‑CoV‑2 and inhibit the viral spike glycoprotein interaction with ACE2. Cell. Mol. Life Sci., 2021, 78, 7777-7794.

Their potential application in cleaning products or nasal sprays, requires additional studies.”

the conclusion has been modified in this sense

The conclusion that the lack of antiviral activity could be attributed to their polarity, which hinders their diffusion through the lipophilic cytoplasmic membrane is quite interesting. Polarity of aminophosphonic acid can be modulated by selective monoesterification protocol (see Molecules 2021, 26, 5637). Products of this reaction 6 of monoester structure are known to efficiently diffuse into the cells what may give the authors compounds with expected antiviral efficacy.

We thank the reviewer for this suggestion and the conclusion was modified with the addition of the suggested reference (reference number 77).

Reviewer 4 Report

The reviewed work concerns 1,3,4-oxadiazole-derived α-aminophosphonates / α-aminophosphonic acids and their antiviral activity. It is a well-written and interesting paper, and worth publishing, but (in my opinion) not in a journal with such a high impact (IF>6.5) as Pharmaceutics. First of all, the synthesis method (although very interesting) has already been described by the authors in 2021 (New J. Chem. 2021, 45, 11327-11335). Now it has been extended with some new examples. The hydrolysis step is rather well-known and should not be considered a novelty.

Other suggestions:
- please check the Dufulin structure (the position of fluorine)
- lines 87-88: you write about: "unprecedented 1,3,4-oxadiazole-derived compounds", but some of these compounds have already been described by you in 2021 (see lines 102-106, New J. Chem. 2021, 45, 11327-11335 ) - that is confusing.

Compounds characterization:
- no physical state (in some cases), no melting points, no IR?
- NMR: 3, 6c - please check the number of signals (13C NMR)
- 5d: are you sure 6J PF > 5JPF?

Supporting:
- page 3: what kind of impurities do we have here (66 ppm)? - they appear often in the other spectra (see also 5e).
- the baselines of 13C NMR spectra (2d, 3, 4, 5b, 5c, 5e, 5f, 6c) are dramatic - what is the problem - this should be corrected.

Author Response

We thank this referee for his/her comments and careful reading of the manuscript.

please check the Dufulin structure (the position of fluorine)

Thank you for the comment, you are perfectly right, the fluorine is in ortho and not in meta as drawn. A corrected figure was added.

lines 87-88: you write about: "unprecedented 1,3,4-oxadiazole-derived compounds", but some of these compounds have already been described by you in 2021 (see lines 102-106, New J. Chem. 2021, 45, 11327-11335 ) - that is confusing.

the term “unprecedented” is now deleted.

no physical state (in some cases), no melting points, no IR?

Melting points of the two compounds displaying the higher biological activities were added. IR measurements were added.

NMR: 3, 6c - please check the number of signals (13C NMR)

The NMR description was checked and corrections were made.

5d: are you sure 6JPF > 5JPF?

Yes, we are sure of these coupling constant, identical constants could be calculated on 31P and 19F spectra.

Supporting: page 3: what kind of impurities do we have here (66 ppm)? - they appear often in the other spectra (see also 5e).

The signal at 66.36 ppm correspond to dioxane, pollution by dioxane occurred for some compounds, we apologize for this inconvenience.

For compound 1 a new 13C NMR spectrum was recorded (supporting page 3).

Supporting: the baselines of 13C NMR spectra (2d, 3, 4, 5b, 5c, 5e, 5f, 6c) are dramatic - what is the problem - this should be corrected.

The spectra are corrected with a different program with a base line correction.

Round 2

Reviewer 4 Report

The manuscript has been corrected according to the instructions. Considering the opinions of other reviewers, I suggest accepting the manuscript in its current form.